# Botanical Antcin K Alleviates High-Fat Damage in Palm Acid Oil-Treated Vascular Endothelial Cells and Macrophages

**DOI:** 10.3390/plants11212812

**Published:** 2022-10-22

**Authors:** Chen-Wen Lu, Ngan Thi Kim Nguyen, Szu-Chuan Shen, Yeh-Bin Wu, Hui-Ju Liang, Chung-Hsin Wu

**Affiliations:** 1School of Life Science, National Taiwan Normal University, Taipei 11677, Taiwan; 2ARJIL Pharmaceuticals LLC, Hsinchu City 30013, Taiwan

**Keywords:** hyperlipidemia, atherosclerosis, oxidative stress, inflammation, antcin K, Kruppel-like factor 4, CD36, vascular endothelial cells, macrophage

## Abstract

Lipid metabolism disorder is the most critical risk factor for atherosclerosis, and the process involves lipid deposition in the arterial intima. In Taiwan, antcin K, an active triterpenoid from the fruiting bodies of *Antrodia camphorata*, has been considered a potential lipid-lowering agent. Despite this, the possible therapeutic mechanisms of antcin K remain unclear. To explore the crucial role of botanical antcin K in reducing atherosclerotic plaque, we used SVEC4-10 vascular endothelial cells and RAW264.7 macrophages with palm acid oil-induced high-fat damage as our cell models. Our results showed through using the DPPH assay that antcin K had excellent free radical scavenging ability. Antcin K treatment can significantly alleviate the high-fat damage and reduce the levels of inflammatory factors of TNF-α and IL-1β in vascular endothelial cells and macrophages, as shown through MTT assay and ELISA. Furthermore, antcin K treatment can effectively enhance migration ability and clear lipid deposition in macrophages, as shown by using cell migration assay and oil red O staining. When stained with immunofluorescence, antcin K was shown to significantly decrease the expression of adhesion molecules of vascular cell adhesion molecule 1 (VCAM-1) in vascular endothelial cells involved in monocyte migration and inflammation. Antcin K not only reduced the expression of the CD36 scavenger receptor but also augmented the expression of Kruppel-like factor 4 (KLF4) transcription factor in macrophages, which inhibits the transformation of macrophages into foam cells underlying the pathological process of atherosclerosis. Taking our findings into account, we suggested that botanical antcin K could have therapeutic potential for the treatment of atherosclerosis.

## 1. Introduction

Atherosclerosis is a chronic inflammatory disease with a series of pathological changes including the following situations: (1) reactive oxygen species (ROS) production and lipid deposition that cause oxidative modification of low-density lipoprotein (LDL), (2) endothelial cell damage that alters endothelial permeability and allows LDL to enter arterial cells, (3) endothelial pro-inflammatory TNF-α and IL-1β production that stimulates circulating monocytes to adhere to endothelial cells and migrate into the subendothelial space, and (4) transformation of macrophages into foam cells. Vascular smooth muscle cells proliferate and migrate to the lesion site. As a result, the formation of atherosclerotic plaques worsens the phenomenon of hardening, stenosis, or even blockages of arteries [1,2,3,4] (Bergheanu et al., 2017, Hansson, 2005, Teng et al., 2020, Ross, 1999). The existence of atherosclerosis, mainly found in coronary arteries, regarding the supply of blood and oxygen to the heart, links to fatal diseases including angina pectoris, myocardial infarction, and sudden death. As is known, hyperlipidemia is one of the most important risk factors for atherosclerosis [5] (Shekelle et al., 1981). In addition, cardiovascular risk factors such as hypercholesterolemia, hypertension, and diabetes can enhance the production of ROS and then increase oxidative modification of lipoproteins and phospholipids, which in turn causes atherosclerosis [6] (Li et al., 2014). As atherosclerosis progresses, plaque rupture and thrombosis induce the narrowing of blood vessels and eventually lead to cardiac dysfunction [7] (Dahlof, 2010).

In the early stage of atherosclerosis development, endothelial damage alters endothelial permeability and allows LDL to enter arterial cells. In response, endothelial cells secrete pro-inflammatory factors that, in turn, stimulate circulating monocytes to adhere to the endothelium and migrate into the subendothelium. Adhesion molecules such as vascular cell adhesion molecule 1 (VCAM-1) are involved in monocyte immigration and inflammation [8] (Hansson. 2001). In human atherosclerotic plaques, the expressions of VCAM-1 and the numbers of macrophages are increased [9] (Nageh et al., 1997). Macrophage activation plays a vital role in the pathological process of atherosclerosis [10,11] (Moore and Tabas, 2011; Moore et al., 2013). It is believed that macrophages originate mainly from circulating monocytes derived from bone marrow [12] (Serbina and Pamer, 2006) or spleen [13] (Robbins et al., 2012). In addition, vascular smooth muscle cells with high proliferative plasticity can be transdifferentiated into macrophages [14] (Chappell et al., 2016). The Kruppel-like factor 4 (KLF4) transcription factor may play a key in regulating macrophage activation that is involved in the formation and progression of atherosclerosis [15,16] (Shankman et al., 2015; Yan et al., 2008). In macrophages, scavenger receptors, especially CD36, are the primary markers that transform lipid-deposited macrophages into foam cells [17] (Kunjathoor et al., 2002). Macrophages have an essential role in lipid regulation because they can engulf lipids, dead cells, and other substances that are considered danger signals in the formation and progression of atherosclerosis [18,19] (Gimbrone Jr. and Garcia-Cardena, 2016; Lusis, 2000). Macrophages can pass through activated vascular endothelial cells, enter the subendothelial cell space to accumulate, and then proliferate into advanced plaques at the lesion site [20] (Robbins et al., 2013). The foam cells induce a series of inflammatory responses, resulting in more lipoprotein retention and persistent chronic inflammation, and finally increase the risk of atherosclerosis [21,22,23] (Hellings et al., 2008; Libby et al., 2000; Merckelbach et al., 2016).

In Taiwan, certain herbal extracts, such as *Antrodia camphorata*, have shown considerable therapeutic potential for atherosclerosis because of anti-inflammatory activity in cellular and preclinical studies [24] (Yang et al., 2017). A previous study disclosed that antcin K, an active triterpenoid extracted from fruiting bodies of *Antrodia camphorate* inhibited the production of pro-inflammatory cytokines and vascular cell adhesion molecule 1 (VCAM-1) in rheumatoid synovial fibroblasts [25,26] (Achudhan et al., 2021; 2022). Moreover, antcin K passed the muscle glucose transporter 4 and AMP-activated protein kinase phosphorylation in glucagon to alleviate hyperlipidemia [27] (Kuo et al., 2016). Although antcin K has been considered to possess therapeutic potential against hyperlipidemia, the mechanism of this has been less extensively studied. As stated, antcin K may act through macrophage targeting strategies such as modulating scavenger receptors, oxidative stress, and inflammatory macrophage to alleviate hyperlipidemia even arteriosclerosis. Our findings provided new insight into the mechanism of botanical antcin K’s therapeutic potential in the treatment of atherosclerosis.

## 2. Results

### 2.1. Chromatographic Fingerprint of Antcin K

Antcin K, an active triterpenoid from the fruiting body of *Antrodia camphorata* [28] was analyzed and confirmed by high-performance liquid chromatography (HPLC) detection methods. *Antrodia camphorata* was identified by external morphology and the marker compound of the plant specimen according to the Taiwan Pharmacopoeia standard. As shown in Figure 1, the bioactive substances present in the HPLC chromatographic fingerprint of *Antrodia camphorata* were antcin K, dehydrosulfurenic acid, sulphurenic acid, versisponic acid D, and dehydroeburicoic acid. We adopted antcin K standard products to prepare a standard solution, and analyzed and compared the standard and sample solution with the same analytical method. The purity of the antcin K standard solution was greater than 90%.

### 2.2. The Free Radical Scavenging Ability of Antcin K

In this study, we mainly study the alleviating effects of botanical antcin K on high-fat damage in palm acid oil-treated vascular endothelial cells and macrophages. First, we examined the effect of antcin K treatment in alleviating oxidation and clearing scavenging free radicals, as shown in Figure 2. The DPPH free radical scavenging ability test is a simple screening test that tests whether an ingredient has antioxidant properties. DPPH (1,1-diphenyl-2-picrylhydrazyl) is a stable free radical. When it dissolves in methanol or ethanol, it will appear blue-violet. When the added component sample can react directly with the DPPH free radical, it will block DPPH in a chain reaction of free radicals. At this time, the color of the blue-purple DPPH solution will turn clear yellow, which means that the added component sample has the ability to capture DPPH free radicals, and the lighter the color, the greater the capture of DPPH free radicals. The stronger the ability, the better the antioxidant ability of this component sample. We used the DPPH assay to compare the quantified DPPH free radical scavenging activity among antcin K treatments at 0–50 μg/mL. We observed that the antcin K treatments at 20 and 50 μg/mL yielded superior antioxidant activity and had good free radical scavenging efficiency (Figure 2A). Compared with standard L-Ascorbic acid, the quantified free radical scavenging efficiency of antcin K is shown in Figure 2B. Antcin K yielded a dose–response manner in free radical scavenging efficiency as follows: 20 μg/mL and 50 μg/mL; antcin K yielded 96.82% and 98.81% of free radicals scavenging efficiency, respectively (Figure 2B, *p* < 0.001). According to Figure 2B, we calculated that IC_50_ of free radical scavenging ability for antcin K treatments was equivalent to 5.76 ± 0.14 μg/mL. DPPH is a stable free radical that turns blue-violet when dissolved in methanol or ethanol. The added sample can react directly with the DPPH free radical. When blocking the chain reaction of DPPH free radicals, the color of the DPPH solution will turn clear yellow. The added sample can capture DPPH free radicals in which the lighter the color, the stronger the ability to capture DPPH free radicals or the higher antioxidant capacity. As suggested in Figure 2B, we choose 20 μg/mL antcin K to study the alleviation of high-fat damage in palm acid oil-treated vascular endothelial cells and macrophages because antcin K at 20 μg/mL should have better antioxidative capacity under high-fat damage palm acid oil treatment. In this study, the lack of antioxidant enzymes and other cellular oxidative parameters such as lipid peroxidation assays can be included in the limitations and future perspectives of the study. In the future, we hope to evaluate cellular oxidative parameters such as lipid peroxidation of botanical antcin K by lipid peroxidation (MDA) assay.

### 2.3. Antcin K Effectively Alleviates Palm Acid Oil-Induced Cytotoxicity in Vascular Endothelial Cells

We examined the effect of palm acid oil treatment on the cytotoxicity of vascular endothelial cells, as shown in Figure 3. Using MTT assay, the vascular endothelial cells treated with palm acid oil (0.25–2 mM) for 24 h induced high-fat damage and reduced cell viability compared with those not treated with PA. We observed that the palm acid oil treatments at 0.25–2 mM for 24 h could induce cytotoxicity in vascular endothelial cells, as shown in Figure 3A. MTT assay is a staining method for detecting the mitochondria of living cells that have been clearly described in a previous report [29] (Rai et al., 2018). This method is based on the fact that mitochondrial succinate-dehydrogenases cleave the tetrazolium ring of MTT and cause a redox reaction to form blue-purple crystals. The purple crystals were dissolved into a purple liquid by the organic solvent of dimethyl sulfoxide (DMSO), and the number of cells was converted by the shade of the generated purple color, thereby measuring the mitochondrial proliferation of cells. 

From Figure 3B, we found that IC_50_ cell viability of vascular endothelial cells for palm acid oil treatment was 2.0 mM. The 75% survival rate of vascular endothelial cells induced by palm acid oil treatment was 0.75 mM. Herein, we selected palm acid oil treatment at 0.75 mM to induce high-fat damage to vascular endothelial cells and RAW264.7 macrophage cells.

Furthermore, we examined the effect of antcin K treatment in alleviating palm acid oil-induced cytotoxicity of vascular endothelial cells, as shown in Figure 4. We found a dose–response relationship between the viability of vascular endothelial cells with palm acid oil-induced high-fat damage and concentrations of antcin K treatments (Figure 4B). Our results showed that the cell viability of vascular endothelial cells decreased to 70.6% after palm acid oil treatment at 0.75 mM concentration. Compared to this, the cell viability of vascular endothelial cells increased to 71.5%, 84.6% (*p* < 0.05), and 109.2% (*p* < 0.01) with antcin K treatments at 10, 20, and 50 μg/mL, respectively. We normalized the cell viability of vascular endothelial cells under sham treatment as 100%. Under this standard, we observed that the change of cell viability of vascular endothelial cells under 10, 20, and 50 μg/mL antcin K treatments was not very obvious without palm acid oil damage but showed a linear increase with doses of antcin K treatments with palm acid oil damage (Figure 4B). Our data suggested that antcin K treatments mainly affect cell growth and reduce cytotoxicity under high-fat damage. Even higher doses of antcin K treatment can promote cell growth greater than 100%.

### 2.4. Antcin K Significantly Alleviates the Palm Acid Oil-Induced Inflammation in Vascular Endothelial Cells

We examined the effect of antcin K treatment in alleviating palm acid oil-induced inflammation of vascular endothelial cells, as shown in Figure 5. Using ELISA, the expressions of TNF-α of vascular endothelial cells were increased from 175.2 to 285.5 pg/mL after 0.75 mM palm acid oil treatment (Figure 5A), and IL-1β of vascular endothelial cells were increased from 69.3 to 86.7 pg/mL after 0.75 mM palm acid oil treatment (Figure 5B). After adding antcin K at 10, 20, and 50 μg/mL, the expressions of TNF-α of vascular endothelial cells were significantly decreased from 285.5 to 178.8 (*p* < 0.01), 180.3 (*p* < 0.01), and 181.0 pg/mL (*p* < 0.01), respectively (Figure 5A); and the expressions of IL-1β of vascular endothelial cells were decreased from 86.6 to 70.2 (*p* < 0.01), 69.3 (*p* < 0.01), and 71.5 pg/mL (*p* < 0.01), respectively (Figure 5B).

### 2.5. Antcin K Significantly Alleviated the Expression of VCAM-1 in Palm Acid Oil-Treated Vascular Endothelial Cells

VCAM-1 adhesion molecule plays a vital role in monocyte immigration, inflammation, and atherosclerosis [9] (Nageh et al., 1997). We examined the effect of antcin K treatment in alleviating expressions of VCAM-1 in palm acid oil-treated vascular endothelial cells, as shown in Figure 6. By immunofluorescence staining, we observed that the expression of VCAM-1 was increased in those palm acid oil-treated vascular endothelial cells without antcin K treatments. However, it was significiantly decreased in those palm acid oil-treated vascular endothelial cells with 20 μg/mL antcin K treatments (Figure 6A). According to our pre-tested data, we choose 20 μg/mL antcin K to study the alleviation of high-fat damage in palm acid oil-treated vascular endothelial cells and macrophages because antcin K at 20 μg/mL has better antioxidative capacity and cell viability under high-fat damage palm acid oil treatment. Compared with those without palm acid oil treatments, the expression of VCAM-1 was significantly increased in the vascular endothelial cells undergoing palm acid oil treatments (Figure 6B, *p* < 0.01). It is noteworthy that 20 μg/mL antcin K treatments reduced the expression of VCAM-1 in those palm acid oil-treated vascular endothelial cells (Figure 6B, *p* < 0.01). 

### 2.6. Antcin K Treatments Effectively Enhanced the Migration Ability of RAW264.7 Macrophages toward Palm Acid Oil-Treated Vascular Endothelial Cells

The effect of antcin K treatment in enhancing the migration ability of RAW264.7 macrophages toward palm acid oil-treated vascular endothelial cells is illustrated in Figure 7. As a result, RAW264.7 macrophages were able to migrate toward palm acid oil-treated vascular endothelial cells. Furthermore, antcin K treatments enhanced RAW264.7 macrophages’ migration to palm acid oil-treated vascular endothelial cells (Figure 7B). Using ELISA, the expression of TNF-α of RAW264.7 macrophages significantly increased from 635.3 to 856.9 pg/mL after 0.75 mM palm acid oil treatment (Figure 7C, *p* < 0.01); however, adding 20 μg/mL antcin K could reduce the expression of TNF-α from 856.9 to 690.7 (Figure 7C, *p* < 0.01). Similarly, the expression of IL-1β of RAW264.7 macrophage cells also increased from 152.3 to 272.3 pg/mL under 0.75 mM palm acid oil (Figure 7C, *p* < 0.05) and decreased from 272.3 to 189.2 after adding 20 μg/mL antcin K (Figure 7C, *p* < 0.05).

### 2.7. Antcin K Effectively Alleviated Lipid Deposition of Palm Acid Oil-Treated Vascular Endothelial Cells

We examined the effect of antcin K treatment in alleviating lipid deposition of palm acid oil-treated vascular endothelial cells using oil red O staining (Figure 8). Although lipid deposition was almost invisible in those vascular endothelial cells that did not receive palm acid oil (Figure 8A), it was obviously seen in palm acid oil-treated vascular endothelial cells (Figure 8B). Furthermore, we applied different concentrations of antcin K in the palm acid oil-treated vascular endothelial cells and found that lipid deposition decreased with concentrations of antcin K treatments (Figure 8B).

### 2.8. Antcin K Significantly Decreased the Expression of CD36 in Palm Oil-Treated Vascular Endothelial Cells

Scavenger receptors expressed in vascular endothelial cells and macrophages, especially CD36, are the primary marker that transforms lipid-deposited macrophages into foam cells. We examined the effect of antcin K treatment concerning the expression of CD36 in palm acid oil-treated vascular endothelial cells, as shown in Figure 9. By immunofluorescence staining, we observed that the expression of CD36 was obviously increased in those palm oil-treated vascular endothelial cells without antcin K treatments but was decreased in those palm oil-treated vascular endothelial cells with 20 μg/mL antcin K treatments (Figure 9A). Compared with those vascular endothelial cells without palm acid oil treatments, the expression of CD36 was significantly increased in those vascular endothelial cells with palm acid oil treatments (Figure 9B, *p* < 0.01). Furthermore, the expression of CD36 was significantly decreased in those palm acid oil-treated vascular endothelial cells after 20 μg/mL antcin K treatments (Figure 9B, *p* < 0.01).

### 2.9. Antcin K Significantly Enhanced the Expression of KLF4 in Palm Oil-Treated Vascular Endothelial Cells

KLF4 plays a crucial role in inhibiting LDL transport and foam cell formation in vascular endothelial cells and macrophages. We examined the effect of antcin K treatment in enhancing the expression of KLF4 in palm acid oil-treated vascular endothelial cells in Figure 10. Immunofluorescence staining showed that KLF4 expression was reduced in palm acid oil-treated vascular endothelial cells without the existence of antcin K treatment; but it was enhanced under 20 μg/mL antcin K treatment (Figure 10A). With palm acid oil treatment, vascular endothelial cells were significantly less likely to express KLF4 than those without palm acid oil treatment (Figure 10B, *p* < 0.01). In addition, 20 μg/mL antcin K significantly increased the expression of KLF4 in palm acid oil-treated vascular endothelial cells (Figure 10B, *p* < 0.01).

## 3. Discussion

The primary goal of this study is to explore the therapeutic potential of botanical antcin K for the treatment of atherosclerosis development. In Taiwan, antcin K, an active triterpenoid from the fruiting bodies of *Antrodia camphorata*, has been proposed as a potential lipid-lowering agent. Our results showed that antcin K had excellent free radical scavenging ability, which can reduce the production of ROS and inhibit oxidative stress (Figure 2). To explore the therapeutic mechanism of antcin K in hyperlipidemia, we used SVEC4-10 vascular endothelial cells and RAW264.7 macrophages with palm acid oil-induced high-fat damage.

As stated, palm acid oil enhances ROS generation in vascular endothelial cells, leading to oxidative stress and causing an oxidative modification of lipoproteins and phospholipids that strengthens the adhesion of monocytes to endothelial cells [30] (Pirillo et al., 2013). In line with previous findings, we observed that vascular endothelial cells were damaged when treated with palm acid oil (Figure 4). Those palm acid oil-treated vascular endothelial cells with high-fat damage enhanced the secretion of pro-inflammatory TNF-α and IL-1β (Figure 5) and significantly increased expressions of adhesion molecules of VCAM-1 (Figure 6). Consistently, the increased secretion of TNF-α and IL-1β could stimulate monocytes to adhere to endothelial cells and migrate into the subendothelial space in the palm acid oil-treated vascular endothelial cells with high-fat damage [31] (Quinn et al., 1987). It has been reported that the secretion of TNF-α and IL-1β may enhance expressions of adhesion molecules of VCAM-1 in vascular endothelial cells [32] (Golias et al., 2007). VCAM-1 was involved in monocyte immigration and inflammation, facilitating the adhesion and transmigration of immune cells [8,33] (Hansson, 2001, Hansson and Libby, 2006).

Antcin K, the main constituent of the fruiting body of *Antrodia camphorata*, has been suggested to display an advantageous therapeutic potential for managing hyperlipidemia [27] (Kuo et al., 2016). Here, we provided new evidence about the possible mechanism of the therapeutic potential of botanical antcin K for the treatment of atherosclerosis. We observed that antcin K can effectively alleviate high-fat damage (Figure 4) and decrease the secretion of pro-inflammatory cytokines of TNF-α and IL-1β (Figure 5) and expressions of adhesion molecules of VCAM-1 (Figure 6) in vascular endothelial cells. In addition, antcin K has excellent free radical scavenging ability, which can reduce the production of ROS and inhibit oxidative stress in damaged endothelial cells. Last but not least, antcin K treatment decreased endothelial permeability to prevent LDL from entering arterial cells in vascular endothelial cell dysfunction. Therefore, anticin K could reduce the chance of atherosclerosis development. As mentioned, the existence of inflammatory cytokines such as TNF-α and IL-1β caused by oxidized LDL-stimulated foam cells brings in impairment of cholesterol efflux in oxidized LDL-loaded macrophages and recruitment of circulating monocytes to atherosclerotic lesions. Consequently, there are more chances of atherosclerotic plaque formation and arterial occlusion. In the present study, palm acid oil-treated RAW264.7 macrophages increased their migration ability (Figure 7B) and secretion of pro-inflammatory cytokines TNF-α and IL-1β (Figure 7C), and increased lipid deposition (Figure 8) and expressions of scavenger receptor of CD36 (Figure 9); by contrast, the expression of transcription factor KLF4 decreased (Figure 10). During the formation of atherosclerosis, macrophages can help to remove oxidized LDL from the intercellular space [10,11] (Moore and Tabas, 2011; Moore et al., 2013). Macrophages may be converted into foam cells by absorbing a large amount of oxidized LDL or impairing oxidized LDL release. Several scavenger receptors are expressed on vascular endothelial cells and macrophages, especially CD36, which is capable of combining with oxidized LDL. With the combination of scavenger receptors of CD36 and oxidized LDL, macrophages efficiently ingest oxidized LDL, accelerate the accumulation of cholesterol, and induce the formation of foam cells in pathological circumstances [17] (Kunjathoor et al., 2002). Unlike the function of CD36, the transcription factor of KLF4 can enhance the expression of various anti-inflammatory and antithrombotic factors including endothelial nitric oxide synthase and thrombomodulin [18,34,35] (Gimbrone et al., 2000; Hamik et al., 2007; Shankman et al., 2015). In addition, KLF4 can reduce TNF-α and IL-1β-induced macrophage activation [16,36] (Feinberg et al., 2005; Yan et al., 2008).

To explore how macrophages can help to target strategies for alleviating atherosclerosis, we examined the effect of antcin K treatment on high-fat damaged vascular endothelial cells. Our results showed that antcin K treatment could enhance migration ability (Figure 7B), reduce the secretion of pro-inflammatory cytokines of TNF-α and IL-1β (Figure 7C), decrease lipid deposition (Figure 8) and the expression of CD36 (Figure 9), and increase the expression of KLF4 (Figure 10). Our results suggest that antcin K treatment can reduce oxidized LDL, which prevents macrophages from secreting inflammatory cytokines such as TNF-α and IL-1β. Consequently, antcin K decreased CD36 expression and increased KLF4 expression, which enhanced oxidized LDL efflux in macrophages, alleviated lipid deposition, and decreased atherosclerotic lesions, thereby reducing atherosclerosis formation.

We summarized our results in Figure 11, demonstrating possible therapeutic mechanisms of antcin K in alleviating the high-fat-damaged vascular endothelial cells and macrophages in blood vessels. When treated with PA, vascular endothelial cells show enhanced ROS generation and inflammatory cytokines of TNF-α and IL-1β and increased expressions of adhesion molecules of VCAM-1. In addition, macrophages show increased lipid deposition and expression of scavenger receptors of CD36 but decreased migration ability and the transcription factor of KLF4 expression. These adverse results led to the oxidative modification of lipoproteins and phospholipids, circulating monocytes adhering to endothelial cells and migrating into subendothelial space, and the transformation of macrophages into foam cells. Notably, anticin K, an active triterpenoid found in the fruiting bodies of *Antrodia camphorata*, can alleviate oxidative modification of lipoproteins and phospholipids in high-fat damaged vascular endothelial cells. Due to its excellent free radical scavenging ability, antcin K prevents monocytes from migrating into subendothelial space by reducing endothelial damage, inflammation secretion of TNF-α and IL-1β, and decreasing expression of VCAM-1. Furthermore, antcin K decreased lipid deposition, enhanced the expression of KLF4, and especially decreased the expression of CD36, effectively preventing macrophages from converting into foam cells underlying high-fat damaged factors. The limitations of this study were a lack of data from animal experiments. In terms of future perspectives, we hope to use an animal model for arteriosclerosis to study botanical antcin K alleviating high-fat damage in high-fat diet-induced atherosclerosis mice in the future.

## 4. Materials and Methods

### 4.1. Determination of Antcin K, an Active Triterpenoid from Antrodia camphorata

As shown in Figure 1, antcin K is an active triterpenoid from the fruiting body of *Antrodia camphorata* [28] (Yang et al., 2022). Dried *Antrodia camphorata* fruiting bodies were ground into a fine powder and extracted with 95% ethanol at ambient temperature by ARJIL Pharmaceuticals. The slurry was filtered, and the filtrate was then concentrated under low pressure to obtain a crude extract. This was followed by suspension in water and extraction with hexane and ether, which were separated using a silica gel column. The ethyl acetate and hexane were then separated by high-performance liquid chromatography (HPLC, SHIMADZU LC 20-A). Further purification yielded antcin K with >90% purity. Conditions of HPLC were summarized as follows: (a) Column: Waters/Sunfire RP18, 5 µm, 150 mm × 4.6 mm ID, (b) Mobile phase: (A) 0.05% TFA aqueous solution, (B) Acetonitrile (ACN), (c) Detection: PDA (λ= 220 nm), (d) Injection volume: 10 µL, (e)Analytic concentration: 0.4 mg/mL, (f) Oven temperature: 30 °C, (g) Run time: 20 min.

### 4.2. Determination of Antioxidant Capacity of Antcin K

The antioxidant capacity of antcin K was examined by 1,1-diphenyl-2-trinitrophenylhydrazyl (DPPH) assay in this study. We formulated DPPH (D9132, Sigma-Aldrich Co., St. Louis, MO, USA) at a concentration of 1.5 mM per 1 mL and added 9 mL of methanol (322415, Sigma-Aldrich Co., St. Louis, MO, USA) to the mixed solution. The 100 μL of DPPH and 100 μL of 10, 20, and 50 μg/mL of test samples were taken, shaken, and mixed evenly, then placed at room temperature for 30 min in the dark. After that, the absorbance was measured at 517 nm using an ultraviolet/visible light spectrometer (Microplate Spectrophotometer, uQuant, Biotek Intruments, Inc., Winooski, VT, USA). Different concentrations of test samples were prepared and compared with the standard L-Ascorbic acid (L-AA) (A5960, Sigma-Aldrich Co., St. Louis, MO, USA). Based on the reduction percentage of the absorbance value observed in the control group, the antioxidant capacity of each test sample to scavenge DPPH free radicals can be judged. After that, the test sample with the most robust ability to scavenge DPPH free radicals was selected and repeated three times. The formula is as follows: DPPH free radical scavenging rate (%) = (1 − (absorption value of experimental group/absorption value of control group)) × 100.

### 4.3. Establishment of SVEC4-10 Vascular Endothelial Cell Model

Vascular endothelial cells (SVEC4-10) were purchased from the Cell Bank of Hsin-Chu Food Industry Development Institute (FIRDI, Cat no: BCRC-60220). SVEC4-10 cells were cultured in high-glucose DMEM (Dulbecco’s Modified Eagle Medium) (Gibco BRL, Grand Island, NY, USA) and supplemented with 10% Fetal Bovine Serum (FBS) (HyClone, Logan, UT, USA), 1.5 g/L sodium bicarbonate, 0.11 g/L sodium pyruvate, 4 mM L-glutamine, 100 U/mL penicillin, and 100 μg/mL streptomycin (Gibco BRL, Grand Island, NY, USA). These cells were grown in a culture medium and maintained in a humidified incubator at 37 °C and 5% CO_2_. Vascular endothelial cells were digested with 0.25% trypsin (Gibco BRL, Grand Island, NY, USA) for approximately 1 min until most cells detached. All cell densities were adjusted to 1 × 10^5^ cells per well for each 24-well culture plate. After vascular endothelial cells were cultured for 24 h, in the hyperlipidemia group, 0.75 mM palm acid oil (CAS 57-10-3-800508; Sigma-Aldrich Co., St. Louis, MO, USA) [37] (Oh et al., 2018) was added, and 10, 20, and 50 μg/mL of antcin K in each experiment was individually added.

### 4.4. Establishment of RAW264.7 Macrophage Cell Model

RAW264.7 macrophage was purchased from the Hsin-Chu Food Industry Development Institute Cell Bank (FIRDI, Cat no: BCRC-60001). The cell culture medium contains high-glucose DMEM (Gibco BRL, Grand Island, NY, USA), 10% FBS, 1.5 g/L sodium bicarbonate, 0.11 g/L sodium pyruvate, 4 mM L-glutamine, 100 U/mL penicillin, and 100 μg/mL streptomycin. These cells were maintained in a humidified incubator at 37 °C and 5% CO_2_. After collecting the cells with a spatula, for each 24-well culture plate, all cell densities were adjusted to 1 × 10^5^ cells per well for migration assays.

### 4.5. Cell Viability Assay

We used a 3-(4,5-Dimethylthiazol-2-yl)-2,5-diphenyltetrazolium bromide (MTT) assay to examine the cell viability of antcin K in this study. A total of 1 × 10^5^ cells/mL SVEC4-10 cells were added to a 24-well plate, and 0.5 mM palm acid oil (Sigma-Aldrich Co., St. Louis, MO, USA) was partially added, and 10, 20, and 50 μg/mL of test samples were added to SVEC4-10 cells, followed by culturing within 24 h. After adding 0.5 mg/mL of MTT solution (M5655, Sigma-Aldrich Co., St. Louis, MO, USA) and culturing for 2 h, the supernatant was removed, and 100 μL/well of DMSO organic solvent was added, followed by a 5 min shaking and then absorbance measurement at 570 nm.

### 4.6. Enzyme-Linked Immunosorbent Assay (ELISA)

We analyzed the content of inflammatory cytokines by ELISA. The supernatant of SVEC4-10 cells with palm acid oil hyperlipidemia treatment was collected after 24 h of culture and added to a 96-well micro-well plate (Thermo Scientific Nunc^®^, Nunc AS, Copenhagen, Denmark). An amount of 100 μL/well of carbonate buffer containing antibodies of 32 ng/mL purified rat anti-mouse TNFα (88-7324; Invitrogen, Carlsbad, CA, USA) or 32 ng/mL purified rat anti-mouse IL-1β (88-7013; Invitrogen, Carlsbad, CA, USA) was added to each well and kept at 4 °C overnight. After that, we washed three times with 0.05% phosphate-buffered saline with Tween 20 (PBST) buffer to remove unbound monoclonal antibodies. Then, we added 200 μL/well of blocking solution to reduce non-specific binding. Next, we washed four times with PBST buffer, added 100 μL/well of 250 × anti-cytokine secondary antibody biotin-conjugated anti-mouse TNF-α and IL-1β antibody, and reacted at room temperature for 1 h. After washing with PBST buffer five times, we added 100 μL/well of avidin-horseradish peroxidase, and the sample was reacted at room temperature for 30 min. Again, samples were washed with PBST buffer six times, reacted with 100 μL/well of tetramethylbenzidine, and kept for 20–30 min at room temperature in the dark for a reaction. The reaction was terminated with 2% H_2_SO_4_, and the absorbance at 450–570 nm was measured.

### 4.7. Cellular Lipid Deposition Assay

We used oil red O staining which was clearly described in a previous report [38] (Luo et al., 2021) to examine cellular lipid deposition. RAW264.7 macrophages were fixed with 2% paraformaldehyde for 10–15 min at room temperature, washed three times with phosphate-buffered saline (PBS) and twice with ddH_2_O, and stained with 5% oil red O dye for 15 min at room temperature, then the supernatant was discarded. Vascular endothelial cells and RAW264.7 macrophages were infiltrated with 60% isopropanol, washed twice with ddH_2_O, and photographed with an optical microscope (Olympus BH2 system light microscope, Olympus Corporation, Tokyo, Japan).

### 4.8. Cellular Migration Assay

We used a cellular migration assay to examine the cellular migration ability of RAW 264.7 macrophages toward vascular endothelial cells with high-fat damage. The cellular migration assay has been clearly described in a previous report [39] (Hwang et al., 2021). Transwells (Corning, New York, NY, USA) with 8.0 μm pore size were used in this experiment for cell migration assays. In total, 1 × 10^5^ RAW 264.7 macrophage, labeled with the red fluorescent dye of 1,1’-Dioctadecyl-6,6’-Di(4-Sulfophenyl) (SP-DilC18(3)), was added to the upper layer of transwell after 48 h. In the lower layer of the chassis, 1 × 10^5^ vascular endothelial cells were seeded in a medium with and without antcin K and palm acid oil groups, which were cultured in a humidified incubator (37 °C and 5% CO_2_). Images were taken with a fluorescence microscope (Leica Microsystems, Wetzlar, Germany), and analysis was performed using Leica Application Suite software (LAS) V4.12 s (Leica Microsystems, Wetzlar, Germany) after 48 h of culture.

### 4.9. Immunofluorescence Staining

We used immunofluorescence staining to examine expressions of KLF4 and CD36. After fixing the cells on the glass slides with 2% paraformaldehyde for 10–15 min at room temperature, the tissues or cells were treated with 0.1% TritonX-100 solution for 15 min at room temperature after washing three times with 1× PBS. After that, the tissues or cells were blocked with 2% BSA for 30 min at room temperature in a humidified dark box to reduce the non-specific binding of antibodies. After washing three times with PBS, the tissues or cells were added 2% BSA to dilute the primary antibody of human anti-mouse VCAM-1 (1:200; sc-13160; Santa Cruz Biotechnology Inc., Dallas, TX, USA), rabbit anti-mouse KLF4 polyclone antibody (membrane protein) (1:200; GTX101508; Genetex, Irvine, CA, USA), or goat anti-mouse CD36 monoclonal antibody (1:200; sc-7309; Santa Cruz Biotechnology Inc.), and then incubated overnight in a humidified dark box at 4 °C. After washing three times with PBS, the secondary antibody anti-mouse IgG-FITC (1:500) diluted with 1% BSA was added, and the cells were reacted with the cells in a humidified dark box for 1 h at room temperature. Note that this step needs to be protected from light at the beginning. Nuclei were stained with 1 μg/mL diamino-2-phenilindole (DAPI) (D3286, Sigma-Aldrich Co., St. Louis, MO, USA) for 10 min at room temperature after washing three times with PBS and mounted with a fluorescent protectant-containing adhesive to reduce fluorescence decay. Fluorescent images were obtained using a Leica DM IRB inverted fluorescence microscope (Leica Microsystems, Wetzlar, Germany), and analysis was performed using Leica Application Suite software (LAS) V4.12 s (Leica Microsystems, Wetzlar, Germany). The Rhodamine fluorescence-labeled cytoskeleton was visualized at 565 nm by exciting cells at 540 nm. Cells are excited at 358 nm and emitted at 461 nm before imaging to visualize blue fluorescence-labeled nuclei.

### 4.10. Statistics and Data Analysis

We used the Student–Newman–Keuls (SNK) multiple comparison post hoc test to compare differences between the groups that were tested. The SNK method is a stepwise multiple comparison method used to identify sample means that are significantly different from each other. In addition, the SNK method uses a stepwise comparison method when comparing sample means. All sample means are sorted in ascending or descending order prior to mean comparison. The largest and smallest sample means are then compared within the largest range. Each group of experiments was repeated at least three times, and all experimental values were expressed as mean values ± standard error of the mean. Differences between the groups were tested using one-way or two-way ANOVA, followed by the SNK multiple comparison post hoc tests. A *p*-value less than 0.05 was considered statistically significant.

## 5. Conclusions

In conclusion, we found that antcin K treatment could effectively alleviate the high-fat damage and reduce the levels of inflammatory factors of TNF-α and IL-1β in vascular endothelial cells and macrophages. Furthermore, antcin K treatment could decrease expressions of VCAM-1 in vascular endothelial cells and the expression of the scavenger receptor of CD36. It is also noteworthy that antcin K could enhance the expression of the transcription factor of KLF4 in macrophages. These mechanisms of modulation by antcin K treatment may prevent the transformation of macrophages into foam cells in the pathological process of atherosclerosis. Based on our findings, botanical antcin K can be a clinically promising therapeutic for preventing and curing atherosclerosis.

## Figures and Tables

**Figure 1 plants-11-02812-f001:**
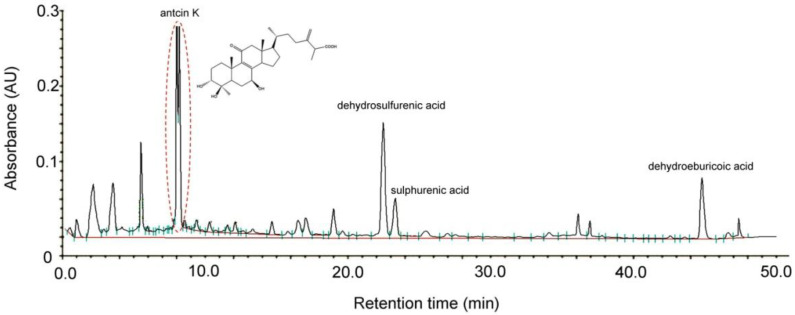
Chromatographic fingerprint analysis by HPLC of *Antrodia camphorata*. The bioactive substances present in HPLC chromatographic fingerprint were antcin K, dehydrosulfurenic acid, sulphurenic acid, versisponic acid D, and dehydroeburicoic acid. Antcin K (3*α*,4*β*,7*β*-tri-hydroxy4*α*-methylergosta-8,24(28)-dien-11-on-26-oic acid, 2) is an active triterpenoid from the fruiting body of *Antrodia camphorata* that was provided from ARJIL Pharmaceuticals LLC. We adopted antcin K standard products to prepare a standard solution, and analyzed and compared the standard and sample solution with the same analytical method. AU, arbitrary perfusion units; HPLC, high-performance liquid chromatography.

**Figure 2 plants-11-02812-f002:**
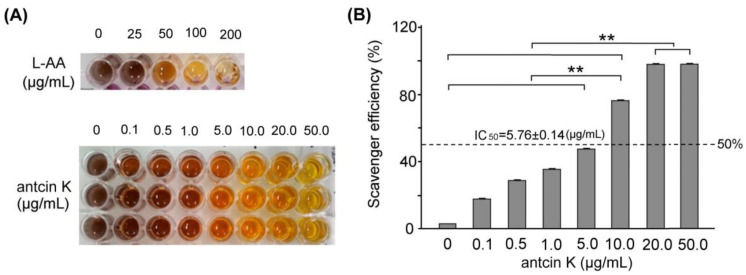
Antioxidative capacity of antcin K. (**A**) DPPH assay using various concentrations of antcin K treatment for 30 min. L-ascorbic acid is a standard antioxidative compound. (**B**) Comparison of the quantified DPPH free radical scavenging activity among antcin K treatments at 0–50 μg/mL. IC_50_ of antcin K was 5.76 ± 0.14 μg/mL (*n*  =  3 for each group; values are presented as mean  ±  SEM, ** *p* < 0.001, one-way ANOVA followed by the Student–Newman–Keuls multiple comparison post hoc test).

**Figure 3 plants-11-02812-f003:**
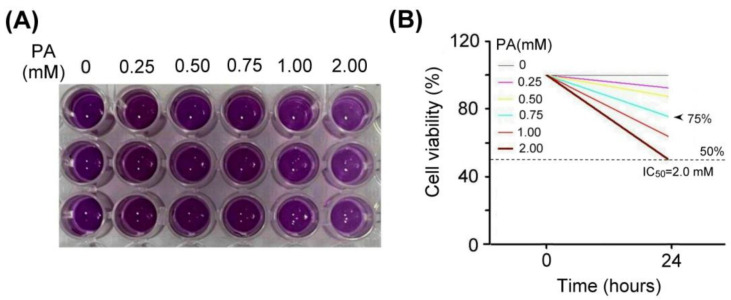
Palm acid oil treatment decreased the cell viability of vascular endothelial cells. (**A**) MTT assay showed the cell viability of vascular endothelial cells under different concentrations of palm acid oil (PA) treatments for 24 h. (**B**) Comparison of the quantified cell viability among vascular endothelial cells with palm acid oil treatments at 0, 0.25, 0.50, 0.75, 1.00, and 2.00 mM. In this experiment, IC_50_ cell viability of vascular endothelial cells for palm acid oil treatment was 2.0 mM. The 75% survival rate of vascular endothelial cells was induced by palm acid oil treatment at 0.75 mM (indicated by the arrow) that was selected to induce high-fat damage of vascular endothelial cells and RAW264.7 macrophage cells as our experimental cell models.

**Figure 4 plants-11-02812-f004:**
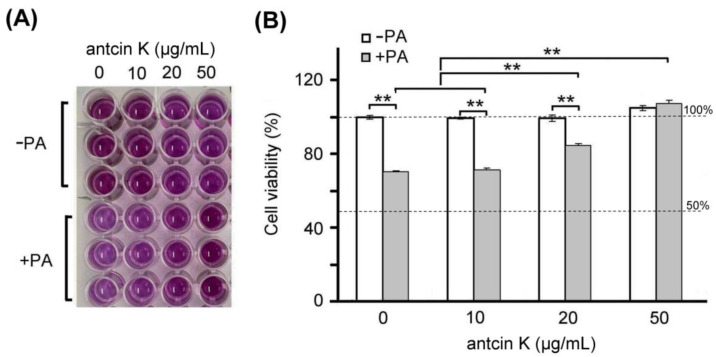
Antcin K treatments effectively alleviated cell viability of palm oil-treated vascular endothelial cells. (**A**) MTT assay showed the cell viability of vascular endothelial cells under different concentrations (0, 10, 20, and 50 μg/mL) of antcin K treatments. (**B**) Comparison of the quantified cell viability among vascular endothelial cells with and without 0.75 mM palm acid oil (PA) treatment and with antcin K treatments at 0, 10, 20, and 50 μg/mL. In this experiment, IC_50_ cell viability of vascular endothelial cells for antcin K treatment was much larger than 50 μg/mL. (*n*  =  3 for each group; values are presented as mean  ±  SEM, ** *p* < 0.01, two-way ANOVA followed by the Student–Newman–Keuls multiple comparison post hoc test).

**Figure 5 plants-11-02812-f005:**
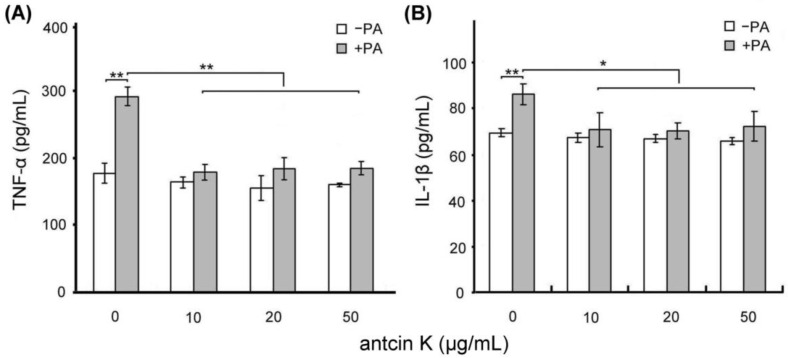
Antcin K treatments effectively alleviated inflammation of palm oil-treated vascular endothelial cells. Comparison of the quantified expressions of the inflammatory markers TNF-α (**A**) and IL-1β (**B**) of vascular endothelial cells with and without 0.75 mM palm acid oil (PA) treatment, and with antcin K treatments at 0, 10, 20, and 50 μg/mL (*n*  =  3 for each group; values are presented as mean  ±  SEM, ** *p* < 0.01, * *p* < 0.05, two-way ANOVA followed by the Student–Newman–Keuls multiple comparison post hoc test).

**Figure 6 plants-11-02812-f006:**
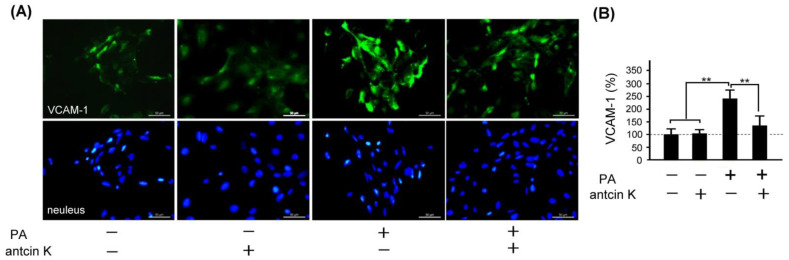
Antcin K treatments effectively reduced the expression of VCAM-1 in palm oil-treated vascular endothelial cells. (**A**) Immunofluorescence staining to examine expressions of VCAM-1 (green) and nuclei (blue, DAPI) in vascular endothelial cells and DAPI with and without palm acid oil (PA) treatments, and with and without antcin K treatments. (**B**) Comparison of the quantified expression of VCAM-1 of vascular endothelial cells with and without 0.75 mM palm acid oil treatment, and with and without 20 μg/mL antcin K treatment (*n*  =  3 for each group; values are presented as mean  ±  SEM, ** *p* < 0.01, one-way ANOVA followed by the Student–Newman–Keuls multiple comparison post hoc test).

**Figure 7 plants-11-02812-f007:**
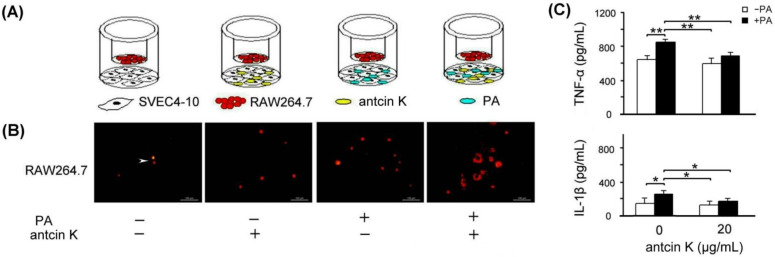
Antcin K treatments effectively enhanced the migration ability of RAW264.7 macrophages toward palm acid oil-treated vascular endothelial cells. (**A**) The cell migration assay for antcin K, which affects the migration ability of RAW264.7 macrophages toward vascular endothelial cells with palm oil-induced high-fat damage. (**B**) Antcin K treatments enhanced the migration ability of RAW264.7 macrophages (indicated by the arrow) toward vascular endothelial cells with palm oil-induced high-fat damage. (**C**) Comparison of the quantified expressions of the TNF-α and IL-1β among RAW264.7 macrophages with and without 0.75 mM palm acid oil (PA) treatment, and with antcin K treatments at 0 and 20 μg/mL (*n*  =  3 for each group; values are presented as mean  ±  SEM, ** *p* < 0.01, * *p* < 0.05, two-way ANOVA followed by the Student–Newman–Keuls multiple comparison post hoc test).

**Figure 8 plants-11-02812-f008:**
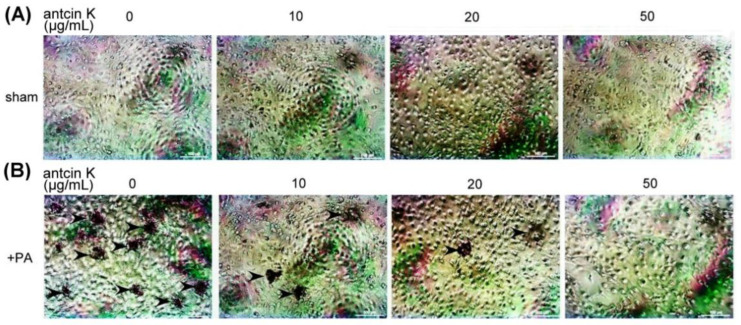
Antcin K treatments effectively decreased the lipid deposition in palm acid oil-treated vascular endothelial cells. Vascular endothelial cells with (**A**) sham and (**B**) palm acid oil (PA) treatments. Cellular lipid depositions were shown by oil red O staining (indicated by the arrow).

**Figure 9 plants-11-02812-f009:**
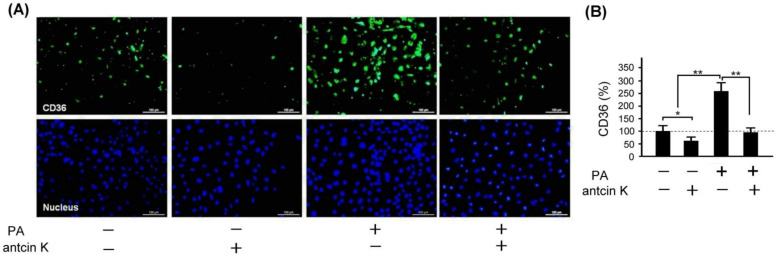
Antcin K treatments effectively reduced the expression of CD36 in palm oil-treated vascular endothelial cells. (**A**) Immunofluorescence staining to examine expressions of CD36 (green) and nuclei (blue, DAPI) in vascular endothelial cells with and without palm acid oil (PA) treatments, and with and without antcin K treatments. (**B**) Comparison of the quantified expression of CD36 of vascular endothelial cells with and without 0.75 mM palm acid oil (PA) treatment, and with and without 20 μg/mL antcin K treatment (*n*  =  3 for each group; values are presented as mean  ±  SEM, * *p* < 0.05, ** *p* < 0.01, one-way ANOVA followed by the Student–Newman–Keuls multiple comparison post hoc test).

**Figure 10 plants-11-02812-f010:**
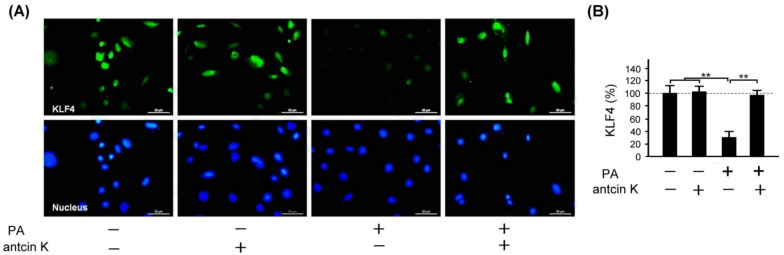
Antcin K treatments effectively enhanced the expression of KLF4 in palm oil-treated vascular endothelial cells. (**A**) Immunofluorescence staining to examine expressions of KLF4 (green) and nuclei (blue, DAPI) in vascular endothelial cells with and without palm acid oil (PA) treatment, and with and without antcin K treatments. (**B**) Comparison of the quantified expression of KLF4 of vascular endothelial cells with and without 0.75 mM palm acid oil treatment, and with and without 20 μg/mL antcin K treatment (*n*  =  3 for each group; values are presented as mean  ±  SEM, ** *p* < 0.01, one-way ANOVA followed by the Student–Newman–Keuls multiple comparison post hoc test).

**Figure 11 plants-11-02812-f011:**
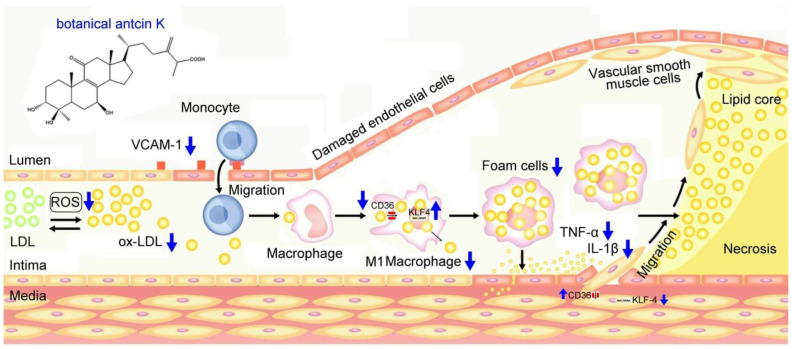
Possible therapeutic mechanisms of antcin K in alleviating the high-fat damage of vascular endothelial cells and macrophages in blood vessels. Antcin K treatment can (1) decrease the expression of VCAM-1 that alleviates circulating monocytes adhering to endothelial cells and migrating into subendothelial space; (2) reduce ROS generation and oxidative stress that alleviate the oxidative modification of lipoproteins and phospholipids; (3) enhance the expression of KLF4 in macrophages and endothelial cells that decrease lipid deposition and prevent macrophages converting into foam cells; (4) decrease the content of TNF-α and IL-1β; and (4) decrease the expression of CD36 that slows transformation of macrophages and endothelial cells to foam cells.

## Data Availability

The data is confidential.

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
