# Peer review of "Botanical Antcin K Alleviates High-Fat Damage in Palm Acid Oil-Treated Vascular Endothelial Cells and Macrophages"

_plants, 2022, doi:10.3390/plants11212812_

Round 1
Reviewer 1 Report
I have reviewed an article: Botanical Antcin K Alleviates High-fat Damage in Palm Acid Oil-treated Vascular Endothelial Cells and Macrophages.
General comment: there are typographical errors so text needs to be reviewed for English language.
Please do not use uncommon abbreviations like PA, YGS etc.
Results section: Is DPPH assay enough to prove antioxidant activity? Usually, DPPH or similar chemical assays are combined with results of antioxidant activity obtained on cells so my suggestion is to conduct addtional experiments on cells used in this work in order to prove antioxidant activity. What was the rationale to test 10 μg/mL, 20 μg/mL, and 50 μg/mL of Antcin K? Please, add explanation in the text. What was the rationale to test concentration range 0.25–2 mM of palmic acid? Please add explanation in the text.
MM section is confusing; the preparation and chemical analysis of traditional herbal medicine comprising seven herbs is described...also, it is described how DPPH assay was conducted, using herbal medicine extract. Yet, in the Results, Antcin K effect was presented. I would like to ask the Authors to add explanation what is the connection between tested extract, Antcin K and Antrodia camphorata.
Furthermore, mutagenicity and western blot experiments conducted on Drosopilla fly is described. There is no mention of Drosophilla in Results and Discussion section. Please, reconsider the text that is written.
Author Response
Dear Reviewer 1:
Many thanks for your recommendation. Please find the third revised manuscript of our submission (Manuscript ID: plants- 1972653) that the title is" Botanical Antcin K Alleviates High-Fat Damage in Palm Acid Oil-Treated Vascular Endothelial Cells and Macrophages" which we re-submitted to Plants. Many thanks for your valuable comments. In accordance with the concerns you have identified, point-for-point responses to your comments and questions are given below by using red description. If there is any problem, please address all correspondence concerning the manuscript to Dr. Chung-Hsin Wu. Best wishes !
Point 1: there are typographical errors so text needs to be reviewed for English language.
Response 1: Many thanks the reviewer's comments. We have corrected typographical errors during revision as possible as we can and checked by a native English-speaking colleague.
Point 2: Please do not use uncommon abbreviations like PA, YGS etc
Response 2: Many thanks the reviewer's comments. We have used full text instead of abbreviations such as PA etc in the text. Please check them in the revised manuscript.
Point 3: Results section: Is DPPH assay enough to prove antioxidant activity? Usually, DPPH or similar chemical assays are combined with results of antioxidant activity obtained on cells so my suggestion is to conduct addtional experiments on cells used in this work in order to prove antioxidant activity. What was the rationale to test 10 μg/mL, 20 μg/mL, and 50 μg/mL of Antcin K? Please, add explanation in the text. What was the rationale to test concentration range 0.25–2 mM of palmic acid? Please add explanation in the text.
Response 3: Many thanks the reviewer's comments. DPPH free radical scavenging ability test is a simple screening test that tests whether an ingredient has antioxidant properties. DPPH (1,1-diphenyl-2-picrylhydrazyl) is a stable free radical. When it dissolves in methanol or ethanol, it will appear blue-violet. When the added component sample can react directly with the DPPH free radical, it will block DPPH. The chain reaction of free radicals. At this time, the color of the blue-purple DPPH solution will turn into clear yellow, which means that the added component sample has the ability to capture DPPH free radicals, and the lighter the color is, it means the capture of DPPH free radicals. The stronger the ability, the better the antioxidant ability of this component sample. This explanatory narrative has been supplemented in the text. In this study, we mainly study alleviating effects of botanical antcin K on high-fat damage in palm acid oil-treated vascular endothelial cells and macrophages, thus we compared the quantified DPPH free radical scavenging activity among antcin K treatments at 0, 10, 20, and 50 μg/mL under palm acid oil treatment or not. Please check them in the revised manuscript.
Point 4: MM section is confusing; the preparation and chemical analysis of traditional herbal medicine comprising seven herbs is described...also, it is described how DPPH assay was conducted, using herbal medicine extract. Yet, in the Results, Antcin K effect was presented. I would like to ask the Authors to add explanation what is the connection between tested extract, Antcin K and Antrodia camphorata.
Response 4: Many thanks the reviewer's comments. We have deleted "prepared standard solutions of different compounds, chlorogenic acid, ferulic acid, liquiritin, glycyrrhizic acid, atractylide III, and ligustilide to identify the bioactive substances present in YGS extract." We added description about Antcin K, an active triterpenoid from the fruiting body of Antrodia camphorata. In this study, we have utilized standard solution of antcin K but not the traditional herbal medicine comprising seven herbs. Please check revised text in "4.1. Determination of antcin K, an active triterpenoid from Antrodia camphorata" in the revised manuscript.
Point 5: Furthermore, mutagenicity and western blot experiments conducted on Drosopilla fly is described. There is no mention of Drosophilla in Results and Discussion section. Please, reconsider the text that is written.
Response 5: Many thanks the reviewer's comments. We have re-written and deleted redundant part that has nothing to do with the content of this manuscript such as Drosopilla fly. Please check them in the revised manuscript.

Reviewer 2 Report
Dear authors,
This study by Lu et al showed interesting results that Antcin K alleviates high-fat damage in palm acid oil-treated vascular endothelial cells and macrophages. However, there are some points that need to be addressed:
Major comments:
(1) While the authors showed and discussed the significant mediation of antcin K after PA-induction, it is noticed that antcin K only treatment also had significant change as compared to normal control (without PA and without antcin K). How would the authors explain this? What is the role of antcin K in a normal (not disease-induced) cell?
(2) Why Student–Newman–Keuls multiple comparison post hoc test was used?
(3) In figure 1, how was scavenger efficiency measured? Is it the same as the inhibitory activity (%) of YGS in terms of DPPH clearance mentioned in Methods? Please clarify in the methods or legend of the figure.
(4) Was the DPPH measurement only done in 30 minutes? Or was it done time-dependently? If only 30 minutes, is there a reason for this exact time measurement? With this result too, was antioxidant capacity considered? Were there antioxidant assays measured?
(5) PA was not defined in the text. Why 24 hours treatment of PA? Was there a pre-test time-dependently?
(6) What happened to other concentrations in figure 7? Why only 20ug/mL was shown. Inconsistent with the other results.
(7) What are the limitations and future perspective of this study?
(8) Methods 4.4-4.7, does this study include animal and drosophila data?
Minor comments:
(9) Abbreviations of words are not fully described when used. What is YGS? Also, Carefully check spellings, spaces and other minor errors (Line 95, 320, 324, others).
(10) In Line 324-325 “and the expression of CD36 (Figure 9), and increase the expression of KLF4 (Figure 10)”- confirm your figure numbering-
Author Response
Dear Reviewer 2:
Many thanks for your recommendation. Please find the third revised manuscript of our submission (Manuscript ID: plants- 1972653) that the title is" Botanical Antcin K Alleviates High-Fat Damage in Palm Acid Oil-Treated Vascular Endothelial Cells and Macrophages" which we re-submitted to Plants. Many thanks for your valuable comments. In accordance with the concerns you have identified, point-for-point responses to your comments and questions are given below by using red description. If there is any problem, please address all correspondence concerning the manuscript to Dr. Chung-Hsin Wu. Best wishes !
Point 1: While the authors showed and discussed the significant mediation of antcin K after PA-induction, it is noticed that antcin K only treatment also had significant change as compared to normal control (without PA and without antcin K). How would the authors explain this? What is the role of antcin K in a normal (not disease-induced) cell?
Response 1: Many thanks the reviewer's comments. We normalized the cell viability of vascular endothelial cells under sham treatment as 100 %. Under this standard, we observed that the change of cell viability of vascular endothelial cells under 10, 20, and 50 μg/mL antcin K treatments was not very obvious without palm acid oil damage, but show a linear increase with dose of antcin K treatments with palm acid oil damage (see Figure 4B). Our data suggested that antcin K treatments mainly affect cell growth and reduces cytotoxicity under with high-fat damage. Even higher dose of antcin K treatment that can promote cell growth and greater than 100%. This explanatory narrative has been supplemented in the text. Please check them in the revised manuscript.
Point 2: Why Student–Newman–Keuls multiple comparison post hoc test was used?
Response 2: Many thanks the reviewer's comments. We used Student–Newman–Keuls (SNK) multiple comparison post hoc test to compare differences between the groups was tested. The SNK method is a stepwise multiple comparison method used to identify sample means that are significantly different from each other. Also, the SNK method uses a stepwise comparison method when comparing sample means. All sample means are sorted in ascending or descending order prior to mean comparison. The largest and smallest sample means are then compared within the largest range. This explanatory narrative has been supplemented in the text. Please check them in the revised manuscript.
Point 3: In figure 1, how was scavenger efficiency measured? Is it the same as the inhibitory activity (%) of YGS in terms of DPPH clearance mentioned in Methods? Please clarify in the methods or legend of the figure.
Response 3: Many thanks the reviewer's comments. We have clarified and supplemented scavenger efficiency measurement of DPPH in "4.2. Determination of antioxidant capacity of antcin K". Please check them in the revised manuscript.
Point 4: Was the DPPH measurement only done in 30 minutes? Or was it done time-dependently? If only 30 minutes, is there a reason for this exact time measurement? With this result too, was antioxidant capacity considered? Were there antioxidant assays measured?
Response 4: Many thanks the reviewer's comments. According to the experimental procedure of DPPH assay, DPPH measurement was done and uniformly determined in exact time of 30 minutes. Based on the reduction percentage of the absorbance value observed in the control group, the antioxidant capacity of each test sample to scavenge DPPH free radicals can be judged. Antioxidant capacity were compared with the standard L-Ascorbic acid (L-AA). This explanatory narrative of DPPH measurement has been supplemented in the text. Please check them in "4.1. Determination of antcin K, an active triterpenoid from Antrodia camphorata" in the revised manuscript.
Point 5: PA was not defined in the text. Why 24 hours treatment of PA? Was there a pre-test time-dependently?
Response 5: Many thanks the reviewer's comments. We have used "palm acid oil" instead of abbreviations of PA in the text. According to other reference (Oh et al., 2018), after vascular endothelial cells were cultured for 24 hours, the 0.75mM palm acid oil was added, and individually added 10, 20, and 50 μg/mL of antcin K in each experiment. Palm acid oil treatment in vascular endothelial cells should be a pre-test time-dependently. Please check them in the revised manuscript.
Point 6: What happened to other concentrations in figure 7? Why only 20ug/mL was shown. Inconsistent with the other results.
Response 6: Many thanks the reviewer's comments. Due to the cumbersome experimental process, we only choose the most effective dose of antcin K to conduct the experiment. According to our pre-tested data, we choose 20 mg/mL antcin K to study the alleviation of high-fat damage in palm acid oil-treated vascular endothelial cells and macrophages because antcin K at 20 mg/mL have better antioxidative capacity and cell viability under high-fat damage palm acid oil treatment. This explanatory narrative has been supplemented in the text. Please check them in the revised manuscript.
Point 7: What are the limitations and future perspective of this study?
Response 7: Many thanks the reviewer's comments. The limitations and future perspective of this study were lack of data from animal experiments in this study. We hope to use an animal model for arteriosclerosis to study botanical antcin K alleviating high-fat damage in high-fat diet-induced atherosclerosis mice in the future. This explanatory narrative has been supplemented in the text. Please check them in the revised manuscript.
Point 8: Methods 4.4-4.7, does this study include animal and drosophila data?
Response 8: Many thanks the reviewer's comments. In this study, we only used SVEC4-10 vascular endothelial cells and RAW264.7 macrophages with palm acid oil-induced high-fat damage as our cell models. We did not use animal and drosophila data in this study. Thus we have deleted those animal and drosophila data that not relevant to this experiment. Please check them in the revised manuscript.
Point 9: Abbreviations of words are not fully described when used. What is YGS? Also, Carefully check spellings, spaces and other minor errors (Line 95, 320, 324, others).
Response 9: Many thanks the reviewer's comments. In the revised manuscript, all abbreviations of words were fully described when first used. We have deleted YGS that not relevant to this experiment. Please check them in the revised manuscript.
Point 10: In Line 324-325 “and the expression of CD36 (Figure 9), and increase the expression of KLF4 (Figure 10)”- confirm your figure numbering.
Response 10: Many thanks the reviewer's comments. We have adjusted the experimental figures graph to match figure number of CD36 (Figure 9) and KLF4 (Figure 10). Please check them in the revised manuscript.

Reviewer 3 Report
In the manuscript entitled “Botanical Antcin K Alleviates High-Fat Damage in Palm Acid Oil-Treated Vascular Endothelial Cells and Macrophages,” authors promote the phytochemical analysis of Antcin K and evaluation potential of protection against damage caused by palm acid oil-induced high-fat model using the vascular endothelial and macrophages cells. All experimental methods are well explained. The methods used are consistent with the literature and corroborate the objectives. The results presented are significant, robust, and supported by other data present in the literature. The conclusions are supported by the results presented. Other Specific comments:
The Chromatographic fingerprint analysis by HPLC shows a 15 peak, and the authors use prepared standard solutions of different compounds, chlorogenic acid, ferulic acid, liquiritin, glycyrrhizic acid, atractylide III, and ligustilide to identify the bioactive substances present in YGS extract. However, if the standard solution of Antcin K doesn’t was utilized, then, as is possible, identify and quantify the presence of Antcin K.
In figure 1, it’s necessary to identify all peaks present in the chromatogram.
The authors need to explain if Antcin K was isolated from YGS extract. If not, how can the authors attribute the observed effects to Antcin K if other compounds exist in the extract?
Calculate the IC50 value to the free radical scavenging ability of Antcin K.
Calculate the IC50 cell viability of vascular endothelial cells for MTT assay. Why was the MTT assay not performed on RAW264.7 macrophage cells?
How can the cell viability of vascular endothelial cells be greater than 100%? I suggest normalizing the data (line 156).
Justify the criteria used to select the 20 μg/mL dose for other experiments; usually, the effective lowest dose is selected.
Why are they not evaluated by cellular oxidative parameters such as lipid peroxidation by MDA assay?
Then, the authors must highlight the real contribution of this paper in relation to that previously published. What is the contribution to state of the art?
The quality of English writing throughout the manuscript needs adjusting. Spacing, punctuation marks, grammar, and spelling errors should be reviewed thoroughly. I found typos during the manuscript. Some sentences in the discussion section look unusual (Generally speaking). The whole manuscript must be evaluated by a native speaker, or a professional of English assistance may be required.
Author Response
Dear Reviewer 2:
Many thanks for your recommendation. Please find the third revised manuscript of our submission (Manuscript ID: plants- 1972653) that the title is" Botanical Antcin K Alleviates High-Fat Damage in Palm Acid Oil-Treated Vascular Endothelial Cells and Macrophages" which we re-submitted to Plants. Many thanks for your valuable comments. In accordance with the concerns you have identified, point-for-point responses to your comments and questions are given below by using red description. If there is any problem, please address all correspondence concerning the manuscript to Dr. Chung-Hsin Wu. Best wishes !
Point 1: The Chromatographic fingerprint analysis by HPLC shows a 15 peak, and the authors use prepared standard solutions of different compounds, chlorogenic acid, ferulic acid, liquiritin, glycyrrhizic acid, atractylide III, and ligustilide to identify the bioactive substances present in YGS extract. However, if the standard solution of Antcin K doesn’t was utilized, then, as is possible, identify and quantify the presence of Antcin K.
Response 1: Many thanks the reviewer's comments. We have deleted "prepared standard solutions of different compounds, chlorogenic acid, ferulic acid, liquiritin, glycyrrhizic acid, atractylide III, and ligustilide to identify the bioactive substances present in YGS extract." As suggested in figure 1, the bioactive substances present in HPLC chromatographic fingerprint of Antrodia camphorata were antcin K, dehydrosulfurenic acid, sulphurenic acid, versisponic acid D, and dehydroeburicoic acid. Antcin K, an active triterpenoid from the fruiting body of Antrodia camphorata. Further purification yielded utilized standard solution of antcin K with >90% purity. In this study, we have utilized standard solution of antcin K. Please check them in 4.1. Determination of antcin K, an active triterpenoid from Antrodia camphorata of the revised manuscript.
Point 2: In figure 1, it’s necessary to identify all peaks present in the chromatogram.
Response 2: Many thanks the reviewer's comments. In figure 1, we have identified larger peaks present in the chromatogram of Antrodia camphorata. The bioactive substances present in antcin K were antcin K, dehydrosulfurenic acid, sulphurenic acid, versisponic acid D, and dehydroeburicoic acid. Please check them in figure 1 of the revised manuscript.
Point 3: The authors need to explain if Antcin K was isolated from YGS extract. If not, how can the authors attribute the observed effects to Antcin K if other compounds exist in the extract?
Response 3: Many thanks the reviewer's comments. Antcin K was isolated from Antrodia camphorata but not from YGS extract. In this study, we utilized standard solution of Antcin K that purity was greater than 90% and provided by ARJIL Pharmaceuticals. Thus we attributed the observed effects to Antcin K but not other compounds. This explanatory narrative has been supplemented in the text. Please check them in the revised manuscript.
Point 4: Calculate the IC50 value to the free radical scavenging ability of Antcin K.
Response 4: Many thanks the reviewer's comments. We have rechecked the effect of antcin K treatment in alleviating oxidation and clearing scavenging free radicals in Figure 2. As suggested in Figure 2B, we calculated that IC50 of free radical scavenging ability for antcin K treatment was equivalent to 5.76±0.14 μg/mL. We have added the IC50 of free radical scavenging ability of antcin K in the text. Please check them in the revised manuscript.
Point 5: Calculate the IC50 cell viability of vascular endothelial cells for MTT assay. Why was the MTT assay not performed on RAW264.7 macrophage cells?
Response 5: Many thanks the reviewer's comments. As suggested in Figure 3B, we calculated that IC50 cell viability of vascular endothelial cells for palm acid oil treatment was 2.0 mM. As suggested in Figure 4B, we calculated that IC50 cell viability of vascular endothelial cells for antcin K treatment was much larger than 50 μg/mL. We have added the IC50 cell viability of vascular endothelial cells for palm acid oil and antcin K treatments in the text. The original experimental design was carried out only for assaying IC50 cell viability of vascular endothelial cells for MTT assay. We will perform MTT assay on RAW264.7 macrophage cells in our further experiment. Please check them in the revised manuscript.
Point 6: How can the cell viability of vascular endothelial cells be greater than 100% ? I suggest normalizing the data (line 156).
Response 6: Many thanks the reviewer's comments. We have normalized the cell viability of vascular endothelial cells under sham treatment as 100 %. Under this standard, we observed the palm acid oil- treated cell viability of vascular endothelial cells under 50 μg/mL antcin K treatments can promote cell growth and even greater than 100%.This explanatory narrative has been supplemented in the text. Please check them in the revised manuscript.
Point 7: Justify the criteria used to select the 20 μg/mL dose for other experiments; usually, the effective lowest dose is selected.
Response 7: Many thanks the reviewer's comments. Due to the cumbersome experimental process, we only choose the most effective dose of antcin K to conduct the experiment. According to our pre-tested data, we choose 20 mg/mL antcin K to study the alleviation of high-fat damage in palm acid oil-treated vascular endothelial cells and macrophages because antcin K at 20 mg/mL have better antioxidative capacity and cell viability under high-fat damage palm acid oil treatment. This explanatory narrative has been supplemented in the text. Please check them in the revised manuscript.
Point 8: Why are they not evaluated by cellular oxidative parameters such as lipid peroxidation by MDA assay?
Response 8: Many thanks the reviewer's comments. In this study, we mainly evaluated cellular oxidative parameters of botanical antcin K by DPPH assay. Due to the limitations of current laboratory equipment and materials, we hope to evaluated cellular oxidative parameters such as lipid peroxidation of botanical antcin K by MDA assay in the future. This explanatory narrative has been supplemented in the text. Please check them in the revised manuscript.
Point 9: Then, the authors must highlight the real contribution of this paper in relation to that previously published. What is the contribution to state of the art?
Response 9: Many thanks the reviewer's comments. We have highlighted the real contribution of this paper in relation to that previously published that botanical antcin K could have therapeutic potential for atherosclerosis. Although antcin K has been considered the therapeutic potential for hyperlipidemia, the mechanism of which has been less extensively studied. Our findings provided new insight into the mechanism of botanical antcin K as the therapeutic potential for atherosclerosis. This explanatory narrative has been supplemented in the text. Please check them in the revised manuscript.
Point 10: The quality of English writing throughout the manuscript needs adjusting. Spacing, punctuation marks, grammar, and spelling errors should be reviewed thoroughly. I found typos during the manuscript. Some sentences in the discussion section look unusual (Generally speaking). The whole manuscript must be evaluated by a native speaker, or a professional of English assistance may be required.
Response 10: Many thanks the reviewer's comments. We have corrected spacing, punctuation marks, grammar, and spelling errors during revision as possible as we can and checked by a native English-speaking colleague. Please check them in the revised manuscript.

Round 2
Reviewer 1 Report
Authors have revised the paper and did necessary corrections according to the reviewers' suggestions. In this form, I think this paper is suitable for publication.
Author Response
Review 1:
Authors have revised the paper and did necessary corrections according to the reviewers' suggestions. In this form, I think this paper is suitable for publication.
Response: Many thanks for review 1 recommendation.
Reviewer 2 Report
The authors addressed the comments well.
Just some minor comments:
*In the response on why only 20ug/mL was used:
"Due to the cumbersome experimental process, we only choose the most effective dose of antcin K to conduct the experiment"
-You may remove this and just use the second part for the explanation. I don't think this is a good reason to use. Experimental process will always be cumbersome because we need to make sure we are reporting the exact findings; the second reason is good enough to justify the ue of 20ug/ml concentration.
*The lack of antioxidant enzymes and other cellular oxidative parameters such as lipid peroxidation assays can be included in the limitation and future perspective of the study.
Author Response
Dear Reviewer 2:
Many thanks for your recommendation. Please find the third revised manuscript of our submission (Manuscript ID: plants- 1972653) that the title is" Botanical Antcin K Alleviates High-Fat Damage in Palm Acid Oil-Treated Vascular Endothelial Cells and Macrophages" which we re-submitted to Plants. Many thanks for your valuable comments. In accordance with the concerns you have identified, point-for-point responses to your comments and questions are given below by using red description. If there is any problem, please address all correspondence concerning the manuscript to Dr. Chung-Hsin Wu. Best wishes !
Point 1: The authors addressed the comments well.
Response: Many thanks for review 2 recommendation.
Point 2: Just some minor comments: "In the response on why only 20ug/mL was used: "Due to the cumbersome experimental process, we only choose the most effective dose of antcin K to conduct the experiment". -You may remove this and just use the second part for the explanation. I don't think this is a good reason to use. Experimental process will always be cumbersome because we need to make sure we are reporting the exact findings; the second reason is good enough to justify the ue of 20ug/ml concentration. *The lack of antioxidant enzymes and other cellular oxidative parameters such as lipid peroxidation assays can be included in the limitation and future perspective of the study.
Response: Many thanks for review 2 comments. As suggested by review 2, we have remove the sentence as follow: In the response on why only 20ug/mL was used: "Due to the cumbersome experimental process, we only choose the most effective dose of antcin K to conduct the experiment", and added the sentence that "The lack of antioxidant enzymes and other cellular oxidative parameters such as lipid peroxidation assays can be included in the limitation and future perspective of the study". Please check them in the revised manuscript.
Reviewer 3 Report
Thanks for the response. But minor explanations are still necessary, please see:
The authors use prepared standard solutions of different compounds, chlorogenic acid, ferulic acid, liquiritin, glycyrrhizic acid, atractylide III, and ligustilide, to identify the bioactive substances present in YGS extract. However, in the HPLC analysis did not observe these compounds on the 15 peaks present in the Chromatographic fingerprint analysis. Please explain.
The authors inform that the standard solution of Antcin K with a purity greater than 90% was provided by ARJIL Pharmaceuticals. It suggested using this to quantify this compound in the YGS extract.
To confirm that the isolated compound is Antcin K , it's necessary to promote the characterization by Infrared spectra, RMN of C13 and H unidimensional and bi-dimensional, and mass spectroscopy.
Author Response
Response to Reviewer 3 Comments
Dear Reviewer 3:
Many thanks for your recommendation. Please find the third revised manuscript of our submission (Manuscript ID: plants- 1972653) that the title is" Botanical Antcin K Alleviates High-Fat Damage in Palm Acid Oil-Treated Vascular Endothelial Cells and Macrophages" which we re-submitted to Plants. Many thanks for your valuable comments. In accordance with the concerns you have identified, point-for-point responses to your comments and questions are given below by using red description. If there is any problem, please address all correspondence concerning the manuscript to Dr. Chung-Hsin Wu. Best wishes !
Point 1: Thanks for the response. But minor explanations are still necessary, please see:
Response 1: Many thanks the reviewer's comments. Point-for-point responses to your comments and questions are given below by using red description.
Point 2: The authors use prepared standard solutions of different compounds, chlorogenic acid, ferulic acid, liquiritin, glycyrrhizic acid, atractylide III, and ligustilide, to identify the bioactive substances present in YGS extract. However, in the HPLC analysis did not observe these compounds on the 15 peaks present in the Chromatographic fingerprint analysis. Please explain.
Response 2: Many thanks the reviewer's comments. We studied effects of antcin K, an active triterpenoid from the fruiting body of Antrodia camphorata. Thus we showed the bioactive substances of Antrodia camphorata but not YGS (previous error description that has been deleted in the manuscript) in HPLC chromatographic fingerprint. As suggested in figure 1, the bioactive substances present in HPLC chromatographic fingerprint of Antrodia camphorata were antcin K, dehydrosulfurenic acid, sulphurenic acid, versisponic acid D, and dehydroeburicoic acid.
Point 3: The authors inform that the standard solution of Antcin K with a purity greater than 90% was provided by ARJIL Pharmaceuticals. It suggested using this to quantify this compound in the YGS extract.
Response 3: Many thanks the reviewer's comments. We have use the standard solution of Antcin K to quantify this compound in the Antrodia camphorata (but not YGS) extract by HPLC chromatographic fingerprint.
Point 4: To confirm that the isolated compound is Antcin K , it's necessary to promote the characterization by Infrared spectra, RMN of C13 and H unidimensional and bi-dimensional, and mass spectroscopy.
Response 4: Many thanks the reviewer's comments. The isolated compound of Antcin K that has been confirmed by ARJIL Pharmaceuticals. Due to the limitations of current laboratory equipment and materials, we will try to isolate compound is Antcin K by Infrared spectra, RMN of C13 and H unidimensional and bi-dimensional, and mass spectroscopy in future perspective of the study.